# ON THE EVALUATION OF GENERATIVE MODELS IN DISTRIBUTED LEARNING TASKS

## ABSTRACT

The evaluation of deep generative models including generative adversarial networks (GANs) and diffusion models has been extensively studied in the literature. While the existing evaluation methods mainly target a centralized learning problem with training data stored by a single client, many applications of generative models concern distributed learning settings, e.g. the federated learning scenario, where training data are collected by and distributed among several clients. In this paper, we study the evaluation of generative models in distributed learning tasks with heterogeneous data distributions. First, we focus on the Fréchet inception distance (FID) and consider the following FID-based aggregate scores over the clients: 1) FID-avg as the mean of clients' individual FID scores, 2) FID-all as the FID distance of the trained model to the collective dataset containing all clients' data. We prove that the model rankings according to the FID-all and FID-avg scores could be inconsistent, which can lead to different optimal generative models according to the two aggregate scores. Next, we consider the kernel inception distance (KID) and similarly define the KID-avg and KID-all aggregations. Unlike the FID case, we prove that KID-all and KID-avg result in the same rankings of generative models. We perform several numerical experiments on standard image datasets and training schemes to support our theoretical findings on the evaluation of generative models in distributed learning problems.

## 1 INTRODUCTION

Deep generative models including diffusion models (Sohl-Dickstein et al., 2015) and generative adversarial networks (GANs) (Goodfellow et al., 2014) have attained impressive results over a wide array of machine learning tasks (Karras et al., 2019; Ho et al., 2020; Ramesh et al., 2022). This success can be attributed to the enormous capacity of multi-layer neural networks in modeling complex distributions of image and text data as well as the intricate design of the training mechanisms in GANs and diffusion models. The promising results of these frameworks have inspired the development of several methodologies for the training and evaluation of generative models in the literature.

While the existing literature on deep generative models has mostly focused on centralized settings with training data stored by a single learner, many modern applications of deep learning algorithms are aimed at *distributed scenarios* where training data are collected by multiple agents in a network. A well-known instance of such a distributed setting is the *federated learning task* (McMahan et al., 2017), where several clients are connected to a server and aim to train a decentralized model through their communications with the server node while preserving the privacy of their collected data. A significant challenge in such distributed learning settings is the heterogeneous data distributions across clients, since the background features of every client could lead to a different data distribution. Especially, in training a deep generative model over a distributed network, the heterogeneity of the clients' distributions could highly impact the performance and evaluation of the trained model.

In this work, we focus on the evaluation of deep generative models in heterogeneous distributed learning settings. Our primary goal is to highlight the challenges of extending standard evaluation metrics for generative models from the centralized setting to the heterogeneous distributed case. To do this, we consider and analyze the following two sensible extensions of an evaluation score: 1) the average score over clients (score-avg), i.e. the mean of the evaluation scores for individual clients, 2) the score with respect to the aggregate distribution of all clients (score-all) considering all the

samples in the network. We consider standard assessment scores and analyze the consistency of model rankings suggested by the described extensions of the score to the distributed learning problem.

While the described score-all aggregation has been commonly used for the evaluation of generative models in the existing literature on distributed and federated generative model learning, we note that the estimation of the score-all metric will be challenging in a real distributed learning scenario. These challenges are mainly due to the privacy and computation constraints in standard distributed learning problems, preventing clients from sharing their observed samples and any revealing statistics of their training data with the rest of the network. On the other hand, the score-avg can be computed more efficiently since it only requires the client-based score value, which needs minor information on the samples and low communication expenses. However, it remains unclear if the original ranking of generative models according to score-all would be preserved under the score-avg assessment, because standard distances used for the evaluation task are non-linear in the input distributions.

In our theoretical analysis, we specifically focus on the Fréchet inception distance (FID) (Heusel et al., 2017) and kernel inception distance (KID) (Bińkowski et al., 2018) scores, which have been widely used in the evaluation of generative models. For the FID score, not only do we prove that the FID-all and FID-avg take different values, but further we show that the model rankings according to the two scores can be different. Indeed, we prove that the generative models with the optimal FID-avg and FID-all scores are different under heterogeneous client distributions, revealing the discrepancy between the two aggregate assessments.

On the other hand, in the case of KID score we prove that the aggregate scores KID-all and KID-avg result in the same rankings of generative models although they could take different values. This result shows the consistency of the KID evaluation between the mean aggregation in KID-avg and the collective data-based aggregation in KID-all. Therefore, our theoretical results suggest the different consistency behaviors of FID and KID scores when aggregated over a heterogeneous distributed learning setting: while averaging individual KID scores results in the same ranking as the KID score with respect to the distribution of collective client's data, the same conclusion does not hold for FID and more generally Wasserstein distance-based evaluation metrics.

Finally, we discuss several numerical results on standard image datasets and generative model architectures which support our theoretical comparison of the aggregate evaluation scores in federated learning problems. Our empirical results suggest that the FID-all and FID-avg aggregations could lead to inconsistent rankings of the trained generative models in standard GANs and diffusion models. In our experiments, we observed that FID-avg often takes higher values when generated samples look sharper. On the other hand, the FID-all score seemed to be higher scores for models with higher diversity in their generated samples. In contrast, KID-all and KID-avg resulted in consistent rankings of the trained models where the evaluated score seems to aggregate both quality and diversity performance. In our numerical experiments, we also evaluated the precision, recall (Sajjadi et al., 2018; Kynkäänniemi et al., 2019), density, and coverage (Naeem et al., 2020) scores under the two introduced aggregations which, except in the case of recall score, could lead to different rankings of the trained model under the discussed score aggregations. In the following, we summarize the main contributions of our study:

- Highlighting the challenges of evaluating generative models in heterogeneous distributed learning settings;
- Analyzing two types of aggregate FID scores in distributed learning scenarios and proving the inconsistencies between the optimal models under the two scores;
- Demonstrating the consistent rankings suggested by the client-wise averaged KID and the aggregate-data-based KID evaluations;
- Presenting numerical results on the evaluation of generative models in distributed settings and empirically supporting the theoretical claims.

## 2 RELATED WORK

A large body of related works (Theis et al., 2016) has focused on the evaluation of generative models in standard centralized learning settings. The existing evaluation scores can be categorized into two general groups: 1) Distance-based metrics defining a distance between the distribution of training data and learnt generative model. The distance between the real and fake distributions is usually

computed after passing the samples through a pre-trained neural net offering a proper embedding of image data. The well-known evaluation scores in this category are the FID (Heusel et al., 2017) and KID (Bińkowski et al., 2018) scores. 2) Quality and diversity-based scores which output a score based on the sharpness and variety of the generated samples. The widely-used evaluation metrics in this category are the Inception score (Salimans et al., 2016), precision and recall metrics (Sajjadi et al., 2018; Kynkäänniemi et al., 2019), and density and coverage scores (Naeem et al., 2020). We note that in this work our goal is not to introduce a new evaluation metric, and our aim is to analyze the extensions of these scores to distributed and federated learning problems under heterogeneous data distributions across clients.

In another set of related works, extensions of generative model training methods including GANs and diffusion models to distributed federated learning have been studied. Rasouli et al. (2020) propose Fed-GAN to train GANs in a federated learning setting. In (Hardy et al., 2019), the gradient from sample generation for the generator is exchanged on the server, while each client possesses a personalized discriminator. According to (Yonetani et al., 2019), different weights are assigned to local discriminators in the non-i.i.d. setting. Conversely, Wu et al. (2022) ensure client privacy by sharing the discriminator across clients, while keeping the generator private. Additionally, Su et al. (2023) explore a dual diffusion paradigm to extend diffusion-based models into the federated learning setting, addressing concerns related to data leakage.

## 3 PRELIMINARIES

### 3.1 DEEP GENERATIVE MODELS

In a deep generative model framework, a neural network generator $G$ is used to map a hidden random vector $\mathbf{Z} \in \mathbb{R}^d$ drawn according to a fixed distribution, e.g. isotropic multivariate Gaussian $\mathcal{N}(\mathbf{0}, \sigma^2 I)$ to a real-like sample $G(\mathbf{Z})$. Several deep learning approaches have been proposed to train such a generator network, including maximum-likelihood-based methods such as variational autoencoder (Kingma & Welling, 2013) and flow-based models (Dinh et al., 2016), generative adversarial networks (GANs) (Goodfellow et al., 2014), and denoising diffusion models (Ho et al., 2020). In our numerical evaluation, we mainly concentrated on the latter two methods, GANs and diffusion models, due to their state-of-the-art performance in computer vision applications.

In GANs, the training of generative models is framed as a min-max game between a generator network $G$ mapping latent vector $\mathbf{Z}$ to a real-like output and a discriminator network $D$ attempting to differentiate $G$'s generated samples from real training data. The GAN game is typically formulated as the following min-max optimization problem where $\boldsymbol{\theta}$, $\boldsymbol{\omega}$ represent the parameters of generator and discriminator neural nets, and $f(G_{\boldsymbol{\theta}}, D_{\boldsymbol{\omega}})$ is the min-max objective representing $D$'s dissimilarity score for the generated and real samples:

$$\min_{\boldsymbol{\theta}} \max_{\boldsymbol{\omega}} f(G_{\boldsymbol{\theta}}, D_{\boldsymbol{\omega}}). \tag{1}$$

The training of GANs in a distributed learning problem aims at solving the above problem via a distributed optimization method. For example, in a federated learning setting where the local clients are connected to a single server node, the training of GAN players can be achieved by a federated min-max optimization algorithm as discussed in the related work section.

In the case of diffusion models, the generative model performs by multi-step denoising of a Gaussian input. The training of this approach is typically done by reversing the denoising process where the training data are turned to a Gaussian input via an iterative addition of independent Gaussian noise vectors. To extend diffusion models in federated learning, we follow the simple FedAvg (McMahan et al., 2017) method and average the locally updated diffusion networks at the server followed by synchronizing the clients with the averaged model.

### 3.2 DISTANCE-BASED EVALUATION OF GENERATIVE MODELS

In order to assess the performance of a generative model, a standard approach is to measure the distance between the distribution of real and generated data. Due to the high-dimensionality of standard image data, the evaluation of image-based generative models is typically performed after passing the data point through a pre-trained Inception model on the ImageNet dataset.

Specifically, a standard distance-based metric is the Fréchet inception distance (FID) defined as the 2-Wasserstein distance between two Gaussian distributions with the mean and covariance parameters of the data distribution $P_X$, denoted by $\boldsymbol{\mu}_X, C_X$, and with the mean and covariance of the generative

model $P_G$, denoted by $\boldsymbol{\mu}_G, C_G$:

$$\text{FID}(P_X, P_G) := \left\| \boldsymbol{\mu}_X - \boldsymbol{\mu}_G \right\|_2^2 + \text{Tr}\left( C_X + C_G - 2\left( C_X C_G \right)^{1/2} \right).$$

Note that we can interpret the FID score as an approximation of the 2-Wasserstein distance given the first and second-order moments of the distributions.

Another widely-used distance-based score for the evaluation of generative models is the kernel inception distance (KID), which measures the maximum mean discrepancy (MMD) between the two distributions which is calculated using a kernel similarity function $k : \mathbb{R}^d \times \mathbb{R}^d \to \mathbb{R}$. Here, the definition of the MMD distance between $P_X$ and $P_G$ based on kernel $k$ follows from

$$\text{KID}(P_X, P_G) := \mathbb{E}_{X, X' \sim P_X}\big[ k(X, X') \big] + \mathbb{E}_{Y, Y' \sim P_G}\big[ k(Y, Y') \big] - 2\,\mathbb{E}_{X \sim P_X, Y \sim P_G}\big[ k(X, Y) \big],$$

where we suppose samples $X, X' \sim P_X$ and $Y, Y' \sim P_G$ are independently drawn. We note that $\text{KID}(P_X, P_G)$ is a non-linear function of input distributions and in the case of a universal kernel function, e.g. Gaussian kernel, is a strictly convex function of the input distributions.

## 4 Evaluation of Generative Models in Distributed Learning Settings

In this section, we discuss two extensions of distance-based evaluation scores from a centralized case to heterogeneous distributed learning settings. In our analysis, we use $\mathcal{D}(P_X, P_G)$ to denote a general distance between data distribution $P_X$ and the generative model $P_G$. For example, $\mathcal{D}$ can be chosen to be the FID score or KID score, which we will analyze later in the section.

In a standard centralized setting, we have only a single distribution $P_X$ for real data. However, the main characteristic of a heterogeneous distributed learning problem is the multiplicity of the involved clients' distribution. Here, we suppose a distributed setting $k$ clients and use $P_{X_1}, \ldots, P_{X_k}$ to denote their underlying distributions, i.e. $P_{X_i}$ stands for the data distribution at client $i$. In addition, we assume that every client $i$ contributes a fraction $0 < \lambda_i < 1$ of the data in the network, that is $\lambda_i = \frac{n_i}{n}$ with $n = \sum_{j=1}^k n_j$ is the total number of samples in the network and $n_i$ is the number of samples at client $i$.

As a result of multiple input distributions, we need to define an aggregate evaluation score that is based on distance measure $\mathcal{D}$. The aggregate distance is supposed to summarize the performance of the generative model $P_G$ in only one score. To do this, we consider and analyze two reasonable ways of defining the aggregate score:

1. **Average Score** $\mathcal{D}_{\text{avg}}$: The score-avg is the mean of the client's individual distance measures, i.e.

$$\mathcal{D}_{\text{avg}}\Big( P_{X_1}, \ldots, P_{X_k} \,; P_G \Big) := \sum_{i=1}^k \lambda_i \mathcal{D}\big( P_{X_i}, P_G \big). \tag{2}$$

2. **Collective-data-based Score** $\mathcal{D}_{\text{all}}$: The score-all with respect to the collective data of the clients is the distance between $P_G$ to the averaged distribution $\widehat{P}_X := \sum_{i=1}^k \lambda_i P_{X_i}$:

$$\mathcal{D}_{\text{all}}\Big( P_{X_1}, \ldots, P_{X_k} \,; P_G \Big) := \mathcal{D}\Big( \widehat{P}_X \,, P_G \Big). \tag{3}$$

In the above, note that $\sum_{i=1}^k \lambda_i P_{X_i}$ is indeed a mixture distribution with $k$ components $P_{X_1}, \ldots, P_{X_k}$ with frequency weights $\lambda_1, \ldots, \lambda_k$. To relate the above aggregate scores, we first observe that when $\mathcal{D}$ is a convex function of the input distributions, which applies to both FID and KID scores, the score-avg $\mathcal{D}_{\text{avg}}$ will upper-bound the score-all $\mathcal{D}_{\text{all}}$:

**Observation 1.** *If $\mathcal{D}(P_X, P_G)$ is a convex function of $P_X$, then*

$$\mathcal{D}_{\text{all}}\Big( P_{X_1}, \ldots, P_{X_k} \,; P_G \Big) \leq \mathcal{D}_{\text{avg}}\Big( P_{X_1}, \ldots, P_{X_k} \,; P_G \Big)$$

**Remark 1.** *The convexity assumption on distance $D$ in the above observation applies to standard divergence scores, including Wasserstein distances, $f$-divergence measures, total variation distance, and the maximum mean discrepancy. Consequently, the result applies to the FID and KID scores.*

While the mentioned observation shows how the two aggregate scores are compared with one another, it does not imply a monotonic relationship between $\mathcal{D}_{\text{all}}\Big( P_{X_1}, \ldots, P_{X_k} \,; P_G \Big)$ and

$\mathcal{D}_{\mathrm{avg}}\Big(P_{X_1},\ldots,P_{X_k}\,;\,P_G\Big)$. Therefore, this observation does not provide a comparison of the ranking of generative models according to the two aggregate scores. In the following subsections, we study this question for the particular FID and KID scores.

## 4.1 Aggregate FID Scores in Distributed Learning

In the case of FID score, we utilize the formulation of FID as the 2-Wassesrtein-distance between the Gaussian-fitted model that leads to a Riemannian geometry. This observation results in the following theorem on FID-all and FID-avg aggregations. We defer the proofs to the Appendix.

**Theorem 1.** *Suppose that $P_{X_1},\ldots,P_{X_k}$ are the clients' distributions with the mean parameters $\boldsymbol{\mu}_1,\ldots,\boldsymbol{\mu}_k$ and covariance matrices $C_1,\ldots,C_k$, respectively, in the semantic space of the Inception net. Then, the followings hold for a generative model $P_G$ with mean $\boldsymbol{\mu}_G$ and covariance $C_G$.*

*1. For FID-all, if we define random $\widehat{X}$ with the average mean $\widehat{\boldsymbol{\mu}} = \sum_{i=1}^{k} \lambda_i \boldsymbol{\mu}_i$ and covariance matrix $\widehat{C} = \sum_{i=1}^{k} \lambda_i \big(C_i + \boldsymbol{\mu}_i \boldsymbol{\mu}_i^\top - \widehat{\boldsymbol{\mu}}\widehat{\boldsymbol{\mu}}^\top\big)$ in the Inception-based semantic space, we have*

$$\mathrm{FID}_{\mathrm{all}}\Big(P_{X_1},\ldots,P_{X_k}\,;\,P_G\Big) \,=\, \mathrm{FID}\big(P_{\widehat{X}},P_G\big).$$

*2. For FID-avg, if we define a random vector $\widetilde{X}$ with the average mean $\widehat{\boldsymbol{\mu}} = \sum_{i=1}^{k} \lambda_i \boldsymbol{\mu}_i$ and covariance matrix $\widetilde{C}$ as the unique solution to $\widetilde{C} = \sum_{i=1}^{k} \lambda_i \big(\widetilde{C}^{1/2} C_i \widetilde{C}^{1/2}\big)^{1/2}$ in the Inception-based semantic space we will have*

$$\mathrm{FID}_{\mathrm{avg}}\Big(P_{X_1},\ldots,P_{X_k}\,;\,P_G\Big) \,=\, \mathrm{FID}\big(P_{\widetilde{X}},P_G\big) + \sum_{i=1}^{k} \lambda_i \mathrm{FID}\big(P_{\widetilde{X}},P_{X_i}\big).$$

*Thus, the FID-all as a function of $G$ is changing monotonically with $\mathrm{FID}\big(P_{\widetilde{X}},P_G\big)$ for defined $\widetilde{X}$.*

**Remark 2.** *In Theorem 1, $\widetilde{C}$ follows from the Wasserstein barycenter of the Gaussian distributions and under the condition that $C_1,\ldots,C_k$ commute, i.e. $C_iC_j = C_jC_i$ for every $i,j$, simplifies to*

$$\widetilde{C} = \Big(\sum_{i=1}^{k} \lambda_i C_i^{1/2}\Big)^2.$$

**Remark 3.** *In Theorem 1, the optimal mean vectors for FID-all and FID-avg aggregations are the same. In contrast, the optimal covariance matrix of FID-avg denoted by $\widetilde{C}$ has no dependence on the choice of $\boldsymbol{\mu}_i$'s, while the optimal covariance matrix of FID-all $\widetilde{C}$ will be affected by the difference between $\boldsymbol{\mu}_i$'s due to the term $\sum_{i=1}^{k} \lambda_i(\boldsymbol{\mu}_i \boldsymbol{\mu}_i^\top - \widehat{\boldsymbol{\mu}}\widehat{\boldsymbol{\mu}}^\top)$. In general, Theorem 1 implies that the gap between FID-all and FID-avg can be written in the following form where* const. *remains constant under different $P_G$'s and $C_G$ is the embedded covariance matrix of $P_G$:*

$$\mathrm{FID}_{\mathrm{avg}}\big(P_{X_1},\ldots,P_{X_k};P_G\big) - \mathrm{FID}_{\mathrm{all}}\big(P_{X_1},\ldots,P_{X_k};P_G\big) = 2\mathrm{Tr}\big((C_G\widehat{C})^{1/2} - (C_G\widetilde{C})^{1/2}\big) + \mathrm{const.}$$

As explained in the above remarks, Theorem 1 shows that the optimal covariance matrices under FID-all and FID-avg could be significantly different in heterogeneous settings with different $\boldsymbol{\mu}_i$'s. Therefore, since the FID-all and FID-avg scores can be interpreted as the distance to covariance matrices $\widehat{C}$ and $\widetilde{C}$, respectively, the rankings suggested by the aggregations will be different if two generators' covariances $C_G$ and $C_{G'}$ have different ordering of distances to $\widehat{C}$ and $\widetilde{C}$.

## 4.2 KID-based Evaluation in Distributed Learning

After showing the possibility of inconsistent rankings by FID-all and FID-avg, we consider the KID score and analyze the consistency of KID-all and KID-avg aggregations. The following theorem proves that unlike the FID-case, the KID-all and KID-avg will result in a consistent ordering of the models and there is a monotonic relationship between the two aggregate scores.

**Theorem 2.** *Consider a kernel function $k : \mathbb{R}^d \times \mathbb{R}^d \to \mathbb{R}$ and the resulting KID score. Then for the clients' distributions $P_{X_1},\ldots,P_{X_k}$ with frequency parameters $\lambda_1,\ldots,\lambda_k$, we will have the following for the average distribution $\widehat{P}_X = \sum_{j=1}^{k} \lambda_j P_{X_j}$:*

$$\mathrm{KID}_{\mathrm{avg}}\Big(P_{X_1},\ldots,P_{X_k}\,;\,P_G\Big) \,=\, \mathrm{KID}_{\mathrm{all}}\Big(P_{X_1},\ldots,P_{X_k}\,;\,P_G\Big) + \sum_{i=1}^{k} \lambda_i \, \mathrm{KID}\big(\widehat{P}_X,P_{X_i}\big),$$

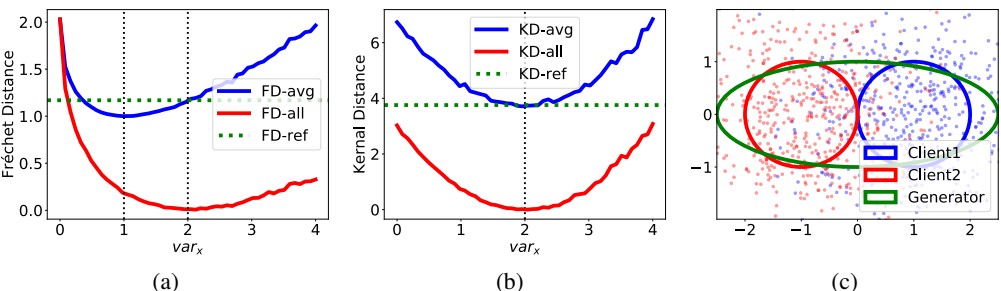

(a)                                (b)                                (c)

Figure 1: Experimental results of Gaussian mixture dataset. (a): The optimal $\mathrm{var}_x$ parameters are different under FD-avg and FD-all evaluations. (b): Distance between KD-avg and KD-all remains the same.(c): the clients' and generator's samples.

*which implies a monotonic relationship between KID-all and KID-avg as a function of $P_G$.*

Therefore, the above theorem shows the consistent rankings implied by the aggregate KID-avg and KID-all scores, as the difference between the scores remains constant while changing the model $P_G$.

## 5 NUMERICAL RESULTS

### 5.1 EVALUATION ON SYNTHETIC GAUSSIAN MIXTURE DATASETS

As discussed in Remark 3, the optimal selection of the covariance matrix differs for the FID-all and FID-avg aggregate scores. To illustrate this distinction, we performed a toy experiment, revealing that FID-avg attains its minimum value when the generator's variance closely approximates that of an individual client, whereas FID-all reaches its minimum value when the generator's variance aligns with that of the aggregated distribution.

**Setup.** Our experimental setup involves two clients, denoted by $C_1$ and $C_2$. $C_1$ possesses a dataset consisting of 50,000 samples drawn from the Gaussian distribution $\mathcal{N}([1,0]^\top, \mathbf{\Sigma})$, while $C_2$ holds a dataset with 50,000 samples drawn from $\mathcal{N}([-1,0]^\top, \mathbf{\Sigma})$, where $\mathbf{\Sigma} = \mathrm{diag}([1,1]^T)$. We introduce a generator, denoted as $G_{\mathrm{var}_x}$, which is parameterized by $\mathrm{var}_x$. $\mathrm{var}_x$ regulates the variance of the generator along the X-axis. Specifically, $G_{\mathrm{var}_x}$ generates 50,000 data points following a Gaussian distribution $\mathcal{N}([0,0]^T, \mathbf{\Sigma_G})$, where $\mathbf{\Sigma_G} = \mathrm{diag}([\mathrm{var}_x, 1]^T)$. The relationship between the two clients and the generator is visually depicted in Figure 1. Additionally, we introduce an "ideal estimator" denoted as $\hat{E} = C_1 \cup C_2$. This ideal estimator possesses the unique ability to replicate the distribution of the training dataset perfectly. We employ the ideal estimator as a reference for our analysis.

**Evaluation Metrics.** We measure the similarity between samples generated by clients and generators using the Fréchet distance (FD), which follows from the Wasserstein-based definition of FID-all and FID-avg without the application of the pre-trained Inception network. We consider the aggregate scores FD-avg and FD-all as defined in Equation (2) and Equation (3). Note that the FD-all for the ideal estimator is zero and we use FD-ref $= \frac{1}{2}\sum_{i=1}^{2} \mathrm{FD}(\hat{E}, C_i)$ as a reference for FD-avg. We also measure the Kernel distance (KD), which follows the definition of KID-all and KID-avg without Inception network. KD-ref is defined for the kernel distance in a similar fashion to FD-ref.

**Results.** By increasing $\mathrm{var}_x$ from 0 to 4, we get a sequence of FD-avg / FD-all pairs and we plot them with the $\mathrm{var}_x$ in Figure 1. Our experimental results highlight the following conclusions. First, we observed that the minimum of FD-all occurs at $\mathrm{var}_x = 2$, while that of FD-avg occurs at $\mathrm{var}_x = 1$, which indicates that the optimal solutions of $\mathrm{var}_x$ to minimize FD-all and FD-avg are inconsistent. In this case, FD-all and FD-avg lead to different rankings of the models with $\mathrm{var}_x = 1$ and $\mathrm{var}_x = 2$. Additionally, we observed that, counterintuitively, the 'ideal estimator' did not reach the minimum average of the Fréchet distances. The distance between KD-avg and KD-all remains the same with the change of $\mathrm{var}_x$ and both of which reach minimum at $\mathrm{var}_x = 2$.

### 5.2 EVALUATION ON REAL IMAGE DATASETS

We evaluated our theoretical results on standard image datasets including CIFAR-10, CIFAR-100 (Krizhevsky et al., 2009), and ImageNet-32 (Deng et al., 2009; Chrabaszcz et al., 2017). In our experiments, we simulated heterogeneous federated learning experiments consisting of non-i.i.d. data

at different clients: for CIFAR-10, we considered 10 clients, each owning 5000 samples exclusively from a single class of the image dataset. Therefore, every client's dataset contains 5000 images having the same label. We put the experiment on CIFAR-100 and ImageNet in Appendix.

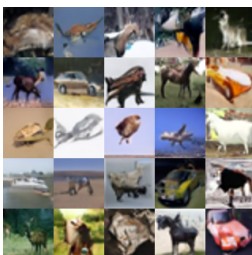 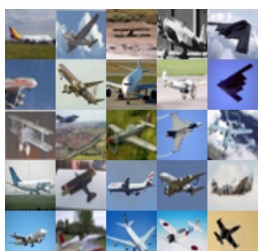

FID-all  =  63.99
FID-avg = 126.05
KID-all  = 0.047
KID-avg = 0.101

FID-all  = 72.13
FID-avg = 119.06
KID-all  = 0.045
KID-avg = 0.098

Figure 2: Left: Images generated by a generative model obtain a lower FID-all. Right: Images from real datasets with class 'plane' obtain a lower FID-avg. FID-avg and FID-all lead to inconsistent rankings, while KID-avg and KID-all result in the same ranking.

**Neural Net-based Generators.** We trained WGAN-GP and DDPM models using the standard training protocols outlined in Gulrajani et al. (2017) and Ho et al. (2020) under the federated CIFAR-10 setting described above. To train the generative models in the federated learning setting, we employed the standard FedAvg approach McMahan et al. (2017). Detailed information regarding the experiment setting can be found in the Appendix. We trained the WGAN-GP generative models multiple times using different random states, and we set different training lengths for every training procedure. We saved the models at different checkpoints every 10 epochs, which is common in training generative models to select the best-performing saved model according to an evaluation metric such as FID and KID.

**Perfect Data-simulating Generators.** In our CIFAR-10 experiments, we also simulated and evaluated an "ideal generator" capable of perfectly replicating all samples belonging to the 'plane' class in CIFAR10. In this scenario, the samples "generated" by the ideal generator exhibit impeccable fidelity but lack diversity since no samples from other categories can be produced.

**FID-based and KID-based Evaluation of Generative Models.** We evaluated the generative models according to FID-all, FID-avg, KID-all, and KID-avg as defined in Section 4. In several cases, we observed that FID-all / FID-avg could assign inconsistent rankings to the generators. Specficially, we computed FID-all and FID-avg for the ideal 'plane'-class-based generator and neural net-based DDPM generators under the distributed CIFAR10 setting. We present some examples generated from the two generators in Figure 2 and report their scores according to the four metrics. The results suggest that FID-avg assigns a considerably higher score to the ideal 'plane'-based generator, whose images preserve perfect details but lack diversity in image categories. Conversely, FID-all assigns a relatively higher value to the DDPM model because its images possess greater diversity. On the other hand, we also observed that KID-avg and KID-all give consistent rankings. Both of them led to the evaluation that the ideal plane generator is slightly better than the DDPM generator.

To further experiment the ranking of generative models according to the discussed aggregate scores, we extracted samples from each class of CIFAR-100 and treated them as the output of one hundred distinct generators, each corresponding to a single class. By assessing these generators on the federated CIFAR-10 dataset, we obtained one hundred pairs of FID-avg / FID-all values, and a subset of these pairs with inconsistent rankings according to FID-all/FID-avg is visualized in the left of Figure 3. The complete set of evaluation results is available in the Appendix. These results further highlight that the rankings provided by FID-all and FID-avg can exhibit inconsistencies in the context of distributed learning. Such inconsistencies could pose a challenge when selecting from a series of checkpoints or model architectures during the training of generative models in distributed learning scenarios, where a distributed computation of FID-all is more challenging than obtaining FID-avg due to privacy considerations.

Regarding the KID-based evaluation, our numerical results suggest that the gap between KID-all and KID-avg remains constant and hence they lead to the same rankings of the geneartive models. Here, we conducted our evaluations on all the geneartive models instances as previously described, and the results are visualized in the left subfigure of Figure 4. These findings reveal that all distinct generators consistently exhibit a uniform gap between KID-avg and KID-all. Consequently, our

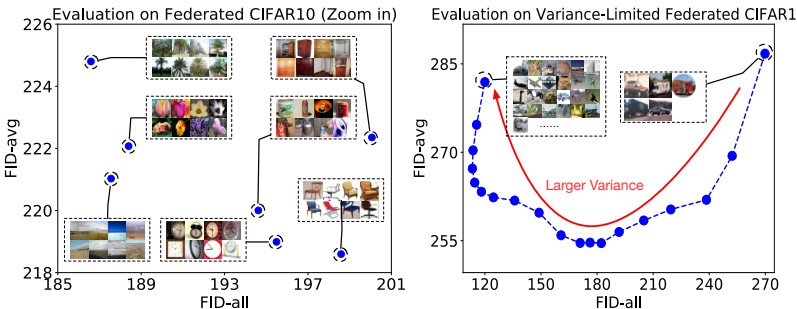

Figure 3: FID-based evaluations on federated CIFAR-10 and variance-limited federated CIFAR-10, FID-avg and FID-all can lead to inconsistent rankings.

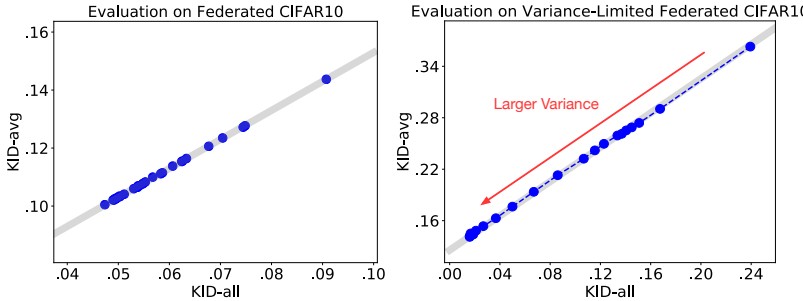

Figure 4: Left: KID evaluations of WGAN-GP checkpoints on federated CIFAR10. Right: KID Evaluations of variance-controlled generators on variance-limited federated CIFAR10.

results indicate that the rankings established by KID-avg consistently align with those of KID-all in distributed learning settings.

### 5.3 Evaluation on Variance-Limited Federated Datasets

In the federated learning literature, it is relatively common that each client possesses only a small portion of the collective dataset, and the data diversity within each client's holdings is significantly constrained. To experiment the effect of such distribution heterogeneity, we simulated and evaluated generative models under *variance-limited federated datasets*. To obtain a variance-limited federated dataset, for each class in the image dataset, we kept only a single image and its K-nearest neighbors. To find the $K$ nearest neighbors, we used the $L_2$-distance in the Inception-V3 2048-dimensional semantic space. This approach effectively mimics scenarios where each client's data has limited variance. We simulated the variance-limited federated learning setting for CIFAR-10, CIFAR-100 and a $32\times32$ version of ImageNet (IN-32) Chrabaszcz et al. (2017). For CIFAR-10 and CIFAR-100, we utilized all the classes in the dataset and for IN-32 we utilized the first 100 classes. We chose $K = 20$ in the experiments. Intuitively, a larger $K$ leads to a more significant intra-client variance.

**Variance-controlled Generators.** To simulate a generator, we initiate the process by randomly selecting a sample from the dataset. We then gather its M-nearest neighbors from the original dataset (w/o federated learning setting). We consider this subset of samples as a set of generated samples generated by a generator denoted by $G_M$. By increasing the value of $M$, we generated a sequence of generators with progressively higher variance values. We tried the $M$ range from 100 to 50000.

**Numerical Results.** We evaluated all the generative models, denoted as $G_M$ with the chosen $M$ values, using the Variance-Limited Federated datasets. The evaluation results on CIFAR10 are in the right subfigure of Figure 3 and Figure 4. Results on ImageNet-32 are illustrated in Figure 5. Our findings reveal a distinct pattern in the behavior of FID-avg and FID-all as generator variance varies while the distance between KID-avg and KID-all remains the same. The result on CIFAR-100 can be found in the Appendix. Our numerical results highlight the impact of the choice of FID-all and FID-avg on model rankings in federated learning settings with limited intra-client variance.

### 5.4 Precision/Recall and Density/Coverege Evaluations

In addition to FID and KID, we followed the definition in Equation (2) and Equation (3) and performed similar experiments to evaluate the consistency between the two aggregate scores for

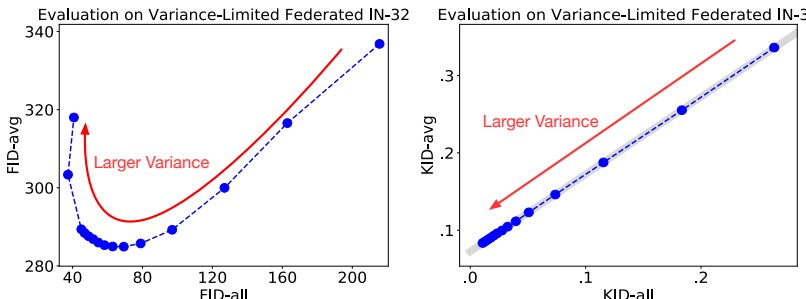

Figure 5: FID and KID-based Evaluations of variance-controlled generators on variance-limited federated ImageNet-32.

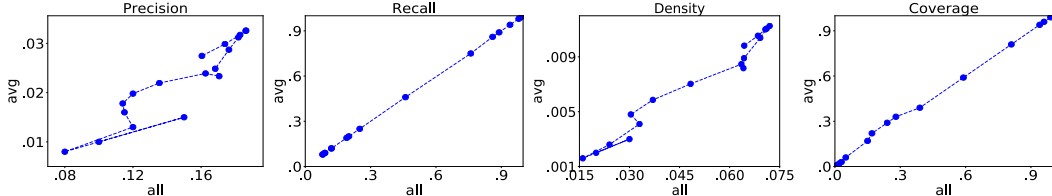

Figure 6: Precision / Recall and Density / Coverage under variance-limited federated CIFAR10.

Precision/Recall (Kynkäänniemi et al., 2019) and Density/Coverage (Naeem et al., 2020). In the case of Precision/Recall, we utilized the official implementation with the number of clusters set to 5. We assessed precision and recall under the variance-limited CIFAR10 setting, with the results presented in Figure 6. The numerical scores indicate that in a heterogeneous data setting, the two aggregate precision scores may not consistently rank the generative models. On the other hand, based on the recall's definition, it can be seen that Recall-all and Recall-avg will always take the same value, since the recall score reduces to an average over the generated data. Our numerical results are also consistent with this observation. For the Density/Coverage evaluations, the numerical results in Figure 6 suggest that while the density-based aggregate scores lead to more consistent rankings under data heterogeneity, both density-all/-avg may still provide inconsistent rankings. On the other hand, the coverage-based evaluations mimic the recall-based evaluation and consistently rank the models.

## 6  CONCLUSION

In this paper, we studied the evaluation of generative models in heterogeneous distributed learning problems where the clients have different distributions. We discussed the challenges of evaluating the overall performance of a trained generative model with only one score and showed the inconsistent rankings of sensible aggregations of standard FID scores in the network. On the other hand, we demonstrated that the same extensions of KID offer the same ranking of generative models. Our theoretical and experimental results indicate that KID-avg can be computed efficiently under the privacy constraints in distributed learning problems, while preserving the KID-all-based ranking of generative models. A possible future direction for our work is to extend the theoretical study to other evaluation criteria such as precision/recall and density/coverage scores. Also, understanding the behavior of the aggregate score using non-arithmetic averaging could be useful for evaluating deep generative models in federated learning contexts.

### LIMITATIONS AND BROADER IMPACT

We note that our numerical study focuses on the applications of generative models to image datasets and the empirical conclusions may not apply to other standard types of data including text and audio data. Regarding the work's broader impact, we note that our analysis could be connected to the fairness evaluation of generative models in distributed learning contexts, as it suggests evaluation metrics for the assessment of diversity in the generated data. The study of fairness and diversity for generative models is required for a principled deployment of generative models in sensitive machine learning applications.

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
