# Supplementary Material of
# On the Evaluation of Generative Models in Distributed Learning Tasks

# 1 Proofs

## 1.1 Proof of Theorem 1

1. Note that according to the definition,

$$\text{FID}_{\text{all}}\Big(P_{X_1}, \ldots, P_{X_k}\,;\, P_G\Big) \;=\; \text{FID}\big(\sum_{i=1}^{k} \lambda_i P_{X_i}, P_G\big).$$

Since the FID score depends only on the the mean and covariance parameters in the Inception-based semantic space, we can replace $\sum_{i=1}^{k} \lambda_i P_{X_i}$ with any other distribution that shares the same mean and covariance parameters, and the FID value will not change. Observe that given mean parameters $\boldsymbol{\mu}_1, \ldots, \boldsymbol{\mu}_k$, the Inception-based mean of $\sum_{i=1}^{k} \lambda_i P_{X_i}$ will be $\widehat{\boldsymbol{\mu}} = \sum_{i=1}^{k} \lambda_i \boldsymbol{\mu}_i$. Therefore, the Inception-based covariance matrix of $\sum_{i=1}^{k} \lambda_i P_{X_i}$ follows from

$$
\begin{aligned}
\sum_{i=1}^{k} \lambda_i \mathbb{E}_{P_i}\big[\big(X_i - \widehat{\boldsymbol{\mu}}\big)\big(X_i - \widehat{\boldsymbol{\mu}}\big)^{\top}\big] &= \sum_{i=1}^{k} \lambda_i \Big[C_i + \big(\boldsymbol{\mu}_i - \widehat{\boldsymbol{\mu}}\big)\big(\boldsymbol{\mu}_i - \widehat{\boldsymbol{\mu}}\big)^{\top}\Big] \\
&= \sum_{i=1}^{k} \lambda_i \Big[C_i + \boldsymbol{\mu}_i \boldsymbol{\mu}_i^{\top}\Big] - \widehat{\boldsymbol{\mu}}\widehat{\boldsymbol{\mu}}^{\top} \\
&= \widehat{C}.
\end{aligned}
$$

Therefore, since we assume $\widehat{X}$ has the Inception-based mean and covariance $\widehat{\boldsymbol{\mu}}$ and $\widehat{C}$, the proof of this part is complete.

2. According to the definition, FID-avg can be written as

$$\text{FID}_{\text{avg}}\Big(P_{X_1}, \ldots, P_{X_k}\,;\, P_G\Big) \;:=\; \sum_{i=1}^{k} \lambda_i \text{FID}\big(P_{X_i}, P_G\big).$$

Therefore, we have

$$
\begin{aligned}
&\text{FID}_{\text{avg}}\Big(P_{X_1}, \ldots, P_{X_k}\,;\, P_G\Big) \\
&\overset{(a)}{=} \sum_{i=1}^{k} \lambda_i W_2^2\Big(\mathcal{N}(\boldsymbol{\mu}_i, C_i), \mathcal{N}(\boldsymbol{\mu}_G, C_G)\Big) \\
&\overset{(b)}{=} \sum_{i=1}^{k} \lambda_i \Big[\|\boldsymbol{\mu}_i - \boldsymbol{\mu}_G\|_2^2 + \text{Tr}\big(C_i + C_G - (C_i C_G)^{1/2}\big)\Big] \\
&= \sum_{i=1}^{k} \Big[\lambda_i \|\boldsymbol{\mu}_i - \boldsymbol{\mu}_G\|_2^2\Big] + \sum_{i=1}^{k} \Big[\lambda_i \text{Tr}\big(C_i + C_G - (C_i C_G)^{1/2}\big)\Big] \\
&\overset{(c)}{=} \|\widehat{\boldsymbol{\mu}} - \boldsymbol{\mu}_G\|_2^2 + \sum_{i=1}^{k} \Big[\lambda_i \|\widehat{\boldsymbol{\mu}} - \boldsymbol{\mu}_i\|_2^2\Big]
\end{aligned}
$$

$$+ \mathrm{Tr}\big(C_G + \widehat{C} - (C_G\widehat{C})^{1/2}\big) + \sum_{i=1}^{k}\Big[\lambda_i\mathrm{Tr}\big(C_i + \widehat{C} - (C_i\widehat{C})^{1/2}\big)\Big]$$

$$= \|\widehat{\boldsymbol{\mu}} - \boldsymbol{\mu}_G\|_2^2 + \mathrm{Tr}\big(C_G + \widehat{C} - (C_G\widehat{C})^{1/2}\big)$$
$$+ \sum_{i=1}^{k}\Big[\lambda_i\|\widehat{\boldsymbol{\mu}} - \boldsymbol{\mu}_i\|_2^2 + \lambda_i\mathrm{Tr}\big(C_i + \widehat{C} - (C_i\widehat{C})^{1/2}\big)\Big]$$

$$= \|\widehat{\boldsymbol{\mu}} - \boldsymbol{\mu}_G\|_2^2 + \mathrm{Tr}\big(C_G + \widehat{C} - (C_G\widehat{C})^{1/2}\big)$$
$$+ \sum_{i=1}^{k}\lambda_i\Big[\|\widehat{\boldsymbol{\mu}} - \boldsymbol{\mu}_i\|_2^2 + \mathrm{Tr}\big(C_i + \widehat{C} - (C_i\widehat{C})^{1/2}\big)\Big]$$

$$\overset{(d)}{=} \mathrm{FID}(P_{\widehat{X}}, P_G) + \sum_{i=1}^{k}\lambda_i\mathrm{FID}(P_{\widehat{X}}, P_{X_i}).$$

In the above, $(a)$ follows from the Wasserstein-based definition of FID distance. $(b)$ comes from the well-known closed-form expression of the 2-Wasserstein distance between Gaussian distributions (Villani et al., 2009). $(c)$ is the result of applying the weighted barycenter of vector $\boldsymbol{\mu}_1, \ldots, \boldsymbol{\mu}_k$ that can be seen to be $\widehat{\boldsymbol{\mu}}$ and the weighted barycenter of positive semi-definite covariance matrices $C_1, \ldots, C_k$ that has been shown to be the unique matrix $\widehat{C}$ that solves the equation $\widetilde{C} = \sum_{i=1}^{k}\lambda_i\big(\widetilde{C}^{1/2}C_i\widetilde{C}^{1/2}\big)^{1/2}$ (Rüschendorf & Uckelmann, 2002; Puccetti et al., 2020). $(d)$ is the direct consequence of the Wasserstein-based definition of the FID distance and the closed-form expression of the 2-Wasserstein distance between Gaussians. Therefore, the proof is complete.

## 1.2 PROOF OF THEOREM 2

To show this theorem, we note that if $\phi(X)$ is the kernel feature map for kernel $k$ used to define the KID distance, i.e. $k(x,y) = \langle\phi(x),\phi(y)\rangle$ is the inner product of the feature maps applied to $x, y$, then it can be seen that the kernel-$k$-based MMD distance can be written as

$$\mathrm{MMD}\big(P_X, P_G\big) := \mathbb{E}_{X,X'\sim P_X}\big[k(X,X')\big] + \mathbb{E}_{Y,Y'\sim P_G}\big[k(Y,Y')\big] - 2\,\mathbb{E}_{X\sim P_X, Y\sim P_G}\big[k(X,Y)\big]$$
$$= \Big\|\mathbb{E}\big[\phi(X)\big] - \mathbb{E}\big[\phi(Y)\big]\Big\|^2.$$

Therefore, following the definition of KID-avg, we can write

$$\mathrm{KID}_{\mathrm{avg}}\big(P_{X_1}, \ldots, P_{X_k}\,;\, P_G\big) := \sum_{i=1}^{k}\lambda_i\mathrm{KID}\big(P_{X_i}, P_G\big)$$

$$= \sum_{i=1}^{k}\lambda_i\mathrm{MMD}_\phi\big(P_{X_i}, P_G\big)$$

$$\overset{(a)}{=} \sum_{i=1}^{k}\lambda_i\Big\|\mathbb{E}\big[\phi(X_i)\big] - \mathbb{E}\big[\phi(G(Z))\big]\Big\|^2$$

$$\overset{(b)}{=} \Big\|\mathbb{E}\big[\phi(\widehat{X})\big] - \mathbb{E}\big[\phi(G(Z))\big]\Big\|^2 + \sum_{i=1}^{k}\Big[\lambda_i\Big\|\mathbb{E}\big[\phi(X_i)\big] - \mathbb{E}\big[\phi(\widehat{X})\big]\Big\|^2\Big]$$

$$\overset{(c)}{=} \mathrm{MMD}_\phi\big(\widehat{P}_X, P_G\big) + \sum_{i=1}^{k}\Big[\lambda_i\mathrm{MMD}_\phi\big(\widehat{P}_X, P_{X_i}\big)\Big]$$

$$\overset{(d)}{=} \mathrm{KID}\big(\widehat{P}_X, P_G\big) + \sum_{i=1}^{k}\Big[\lambda_i\mathrm{KID}\big(\widehat{P}_X, P_{X_i}\big)\Big]$$

$$\overset{(e)}{=} \mathrm{KID}_{\mathrm{all}}\big(P_{X_1}, \ldots, P_{X_k}\,;\, P_G\big) + \sum_{i=1}^{k}\lambda_i\mathrm{KID}\big(\widehat{P}_X, P_{X_i}\big).$$

In the above, $(a)$ and $(c)$ follow from the feature-map-based formulation of the MMD distance. $(b)$ is the consequence of the fact that $\|\cdot\|$ is the norm in a reproducing kernel Hilbert space and for $\widehat{X}$ distributed as $\widehat{P}_X = \sum_{i=1}^{k} \lambda P_{X_i}$ we know that $\mathbb{E}\big[\phi(\widehat{X})\big]$ is the weighted barycenter of the individual mean vectors $\mathbb{E}\big[\phi(X_1)\big], \ldots, \mathbb{E}\big[\phi(X_k)\big]$. $(d)$ is based on the definition of KID. Finally, $(e)$ follows from the definition of KID-all, which completes the proof.

## 2 TRAINING DETAILS

We have trained WGAN-GP Salimans et al. (2016) and DDPM Ho et al. (2020) in a federated learning setting by utilizing FedAvg approach McMahan et al. (2017). The experiment protocols for WGAN-GP and DDPM are copied from original works. The communication interval of FedAvg is set as 160 iterations for both WGAN-GP and DDPM. We have tried different communication intervals for both models. The communication frequency will affect model performance but have no influence on the conclusions in the main part of our paper.

## 3 EXTRA EXPERIMENT RESULTS

In this section, we show the some extra experiment results.

### 3.1 EVALUATE CIFAR100 ON FEDERATED IMAGENET-32

We expand the evaluation of CIFAR100 to Federated ImageNet-32 dataset. Similarly, we extracted samples from each class of CIFAR-100 and treated them as the output of one hundred distinct generators, each corresponding to a single class. We also keep the first one hundred classes of ImageNet-32 and simulate one hundred clients. Each client hold all images ($\sim$1300) from a single class. We evaluate all the generators on Federated ImageNet-32 and the result is shown in Figure 1. The ranks provided by FID-avg and FID-all is inconsistent in a much more complex distributed learning setting.

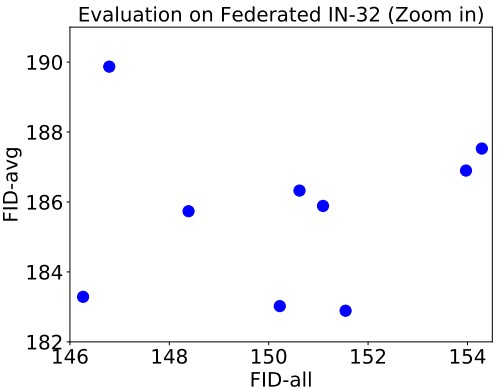

Figure 1: Evaluate CIFAR100 on Federated ImageNet-32.

### 3.2 EVALUATION OF WELL-TRAINED GENERATORS IN DISTRIBUTED LEARNING CONTEXTS

Here, we extend our numerical analysis to a federated learning scenario with realistic generators. We use the widely used StyleGAN2-ADA(Karras et al., 2020) generator, and download an FFHQ pre-trained model from the StyleGAN2-ADA's GitHub repository. The generated images have the size $1024 \times 1024$. Similar to the simulated generator setting in the main body of this paper, we synthesized a sequence of variance-controlled generators by applying the standard truncation technique (Kynkäänniemi et al., 2019) on the random noise $\mathbf{z}$. The truncation parameter $\tau$ varies from $0.01$ to $1.0$. A greater truncation parameter $\tau$ leads to higher diversity in generated samples. We illustrate how the truncation parameter affects the samples in Figure 2. For each truncation $\tau$, we generate 5000 samples from the model for the evaluation.

Furthermore, to simulate a distributed learning setting where each client only possesses a small set of variance-limited samples, we synthesized 100 clients. Each of the clients hold 100 images

synthesized by the model with truncation $\tau = 0.25$. The images within each client look similar, while images across clients look highly different, as shown in Figure 3,

Similarly, we evaluate the models based on the two aggregations of FID and KID scores in this setting. The numerical results are shown in Figure 4. The FID-avg vs. FID-all plot leads to a U-shape curve, while the gap between KID-avg and KID-all remains constant for different generators. We further show the relation between the ranking based on FID and KID aggregate scores in Figure 5, which looks similar to the results we presented on the Mini-ImageNet dataset (Figure 6) in the text.

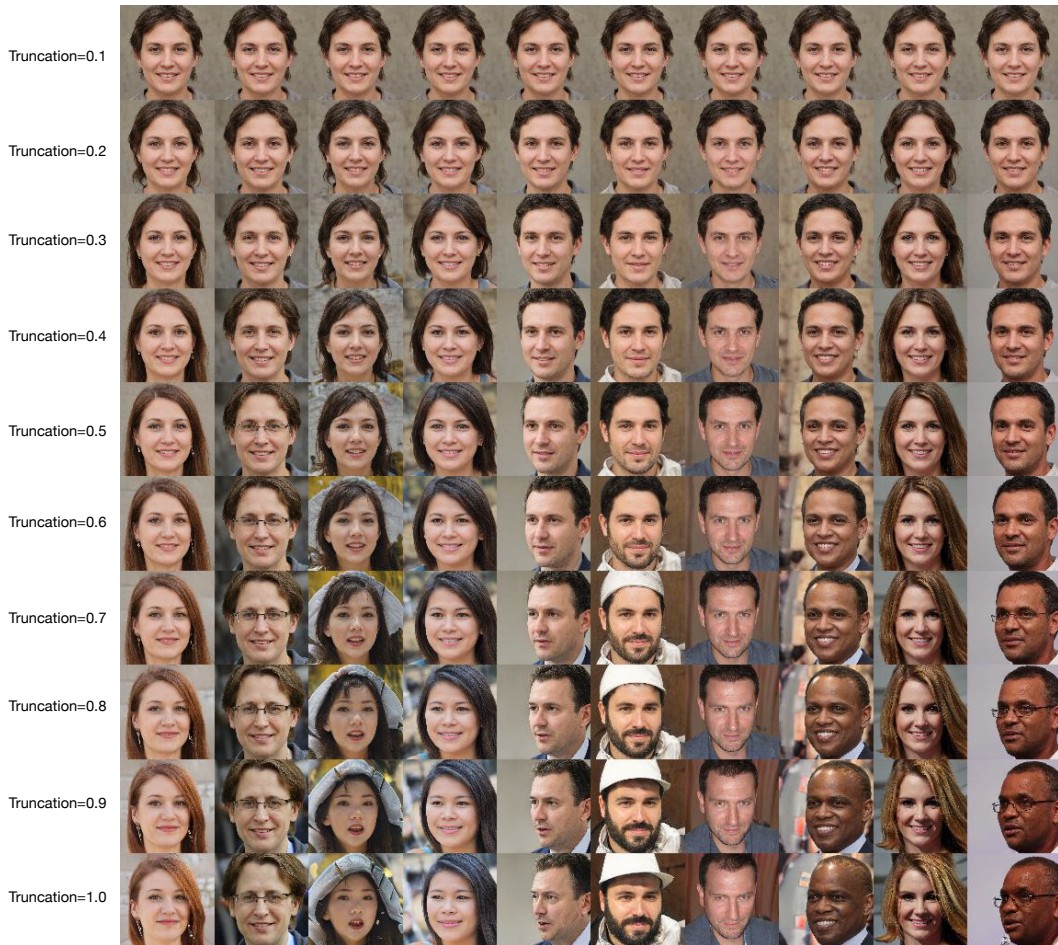

Figure 2: Illustration of randomly-generated samples from the variance-controlled generators.

### 3.3 KID-BASED EVALUATION USING GAUSSIAN KERNEL KID DISTANCE

In our numerical discussion in the text, we utilized the standard implementation of KID measurement from data with a polynomial kernel, $k(\mathbf{x}, \mathbf{y}) = \left(\frac{1}{d}\mathbf{x}^{\mathbf{T}}\mathbf{y} + 1\right)^{\mathbf{3}}$, where $d$ is the dimension of feature vector. We note that our theoretical finding on the evaluation consistency under KID-all and KID-avg applies to every kernel similarity function. We redid the experiment in Fig.2 with a Gaussian RBF kernel $k_\sigma^{\mathrm{rbf}}(\mathbf{x}, \mathbf{y}) = \exp\left(-\frac{1}{2\sigma^2}\|\mathbf{x} - \mathbf{y}\|^{\mathbf{2}}\right)$ as formulated in Bińkowski et al. (2018), where we chose $\sigma = \sqrt{d}$ in the experiments. For randomly sampled images from a CIFAR10 pre-trained diffusion model in the experiment, KID$^{\mathrm{rbf}}$-all gives $4.277e^{-3}$ while KID$^{\mathrm{rbf}}$-avg gives $4.295e^{-3}$. And for the airplane images in CIFAR10, KID$^{\mathrm{rbf}}$-all gives $4.283e^{-3}$ while KID$^{\mathrm{rbf}}$-avg gives $4.301e^{-3}$. The results indicate that for Gaussian RBF kernel $k_\sigma^{\mathrm{rbf}}$, KID$^{\mathrm{rbf}}$-all and KID$^{\mathrm{rbf}}$-avg still gives consistent results. In this case, the KID$^{\mathrm{rbf}}$-based evaluation suggests the images sampled from the diffusion model have higher quality than the set of airplane images in the CIFAR10 dataset.

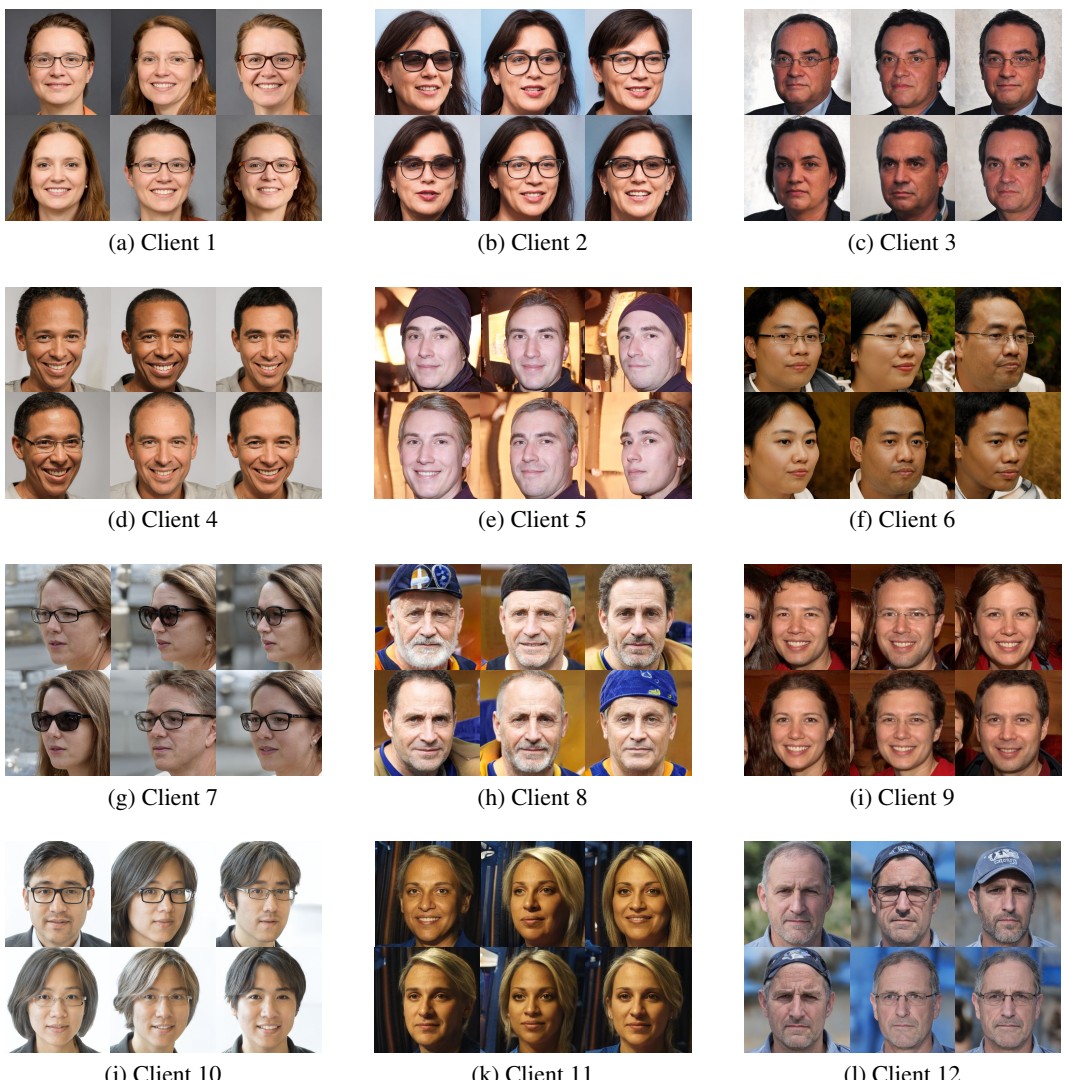

(a) Client 1    (b) Client 2    (c) Client 3

(d) Client 4    (e) Client 5    (f) Client 6

(g) Client 7    (h) Client 8    (i) Client 9

(j) Client 10    (k) Client 11    (l) Client 12

Figure 3: Illustration of random samples from randomly selected variance-limited clients.

### 3.4 EVALUATION OF SYNTHETIC GAUSSIAN MIXTURE DATA WITH THE LOG-LIKELIHOOD SCORE

We also evaluated the synthetic Gaussian mixture dataset mentioned in Section 5.1 with the standard log-likelihood (LL) score. In this experiment, we note that we have access to the probability density functions (PDF) of the simulated generator. We utilized the generator $G_{var_x}$ described in the main text and performed the evaluation over the parameter $var_x$ in the range $[0, 40]$. As can be shown in the general case, LL-avg and LL-all led to the same value for every evaluated model. As shown in Figure 7a, they reached their maximum value at $var_x = 2$. On the other hand, we set a new generator $G_{mean_x}$ generating samples according to $\mathcal{N}([mean_x, 0]^\top, \Sigma)$, where $\Sigma = \mathrm{diag}([2, 1]^T)$. We gradually increased $mean_x$ from -2 to 2 and plotted LL-avg, LL-all, and LL-ref in Figure 7b.

### 3.5 RESULTS ON VARIANCE-LIMITED FEDERATED CIFAR100

Similar to the experiments on CIFAR10 and IN-32, we have also applied the variance-limited federated dataset setting to CIFAR100. We keeps K=20 images in each class. For variance-controlled generators, we select a sample from original CIFAR100 and gather the M-nearset neighbors. The range of M keeps the same with that in the main part of this paper. We show the results in Figure 8.

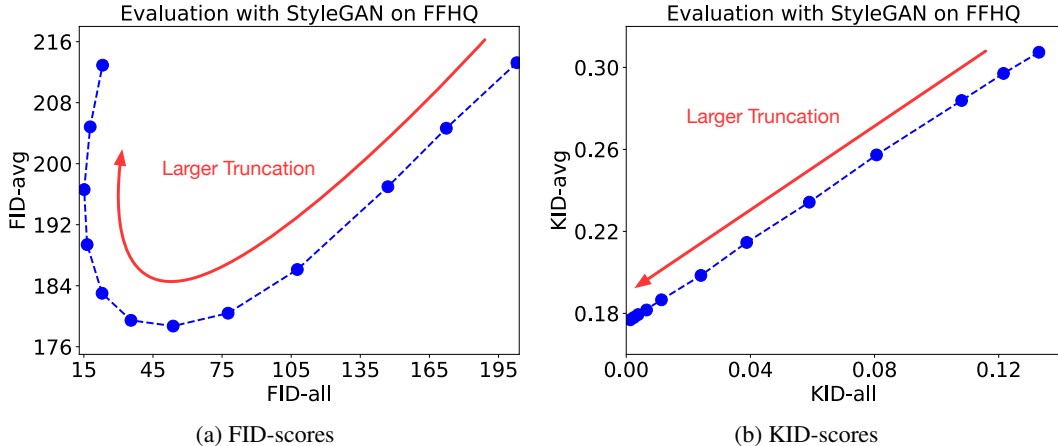

(a) FID-scores

(b) KID-scores

Figure 4: Evaluation on FFHQ with StyleGAN.

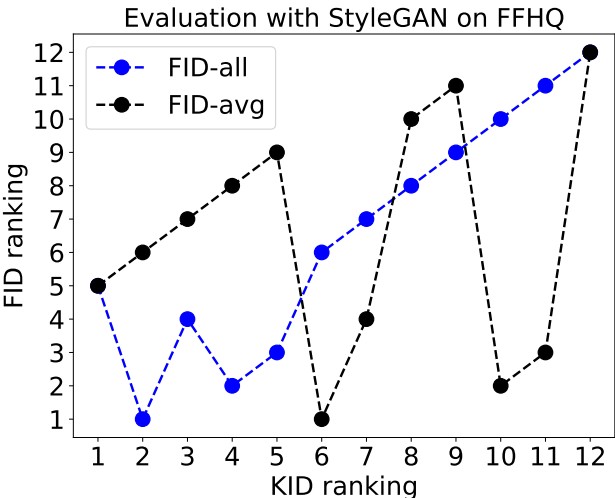

Figure 5: Comparing FID-based and KID-based rankings of truncated StyleGAN-based generative models with different truncation parameters. The lower the rank is, the better.

The results still support our main claims: FID-avg and FID-all gives inconsistent results while KID-avg and KID-all give the same.

### 3.6 GENERAL 1-WASSERSTEIN-DISTANCE EVALUATION METRICS

Let $\mathbb{P}_g$ and $\mathbb{P}_t$ represent the distribution of generated set and training set. The Wasserstain-1 distance between $\mathbb{P}_g$ and $\mathbb{P}_t$ is,

$$W(\mathbb{P}_g, \mathbb{P}_t) = \inf_{\lambda \in \Pi(\mathbb{P}_g, \mathbb{P}_t)} \mathbb{E}_{(x,y) \sim \lambda}[\|x - y\|], \tag{1}$$

where $\Pi(\mathbb{P}_g, \mathbb{P}_t)$ denotes the set of all joint distribution $\lambda(x, y)$ whose marginal distribution are respectively $\mathbb{P}_g$ and $\mathbb{P}_t$. However, the direct estimation of $W(\mathbb{P}_g, \mathbb{P}_t)$ is highly intractable. On the other hand, the Kantorovich-Rubinstein duality Villani et al. (2009) gives,

$$W(\mathbb{P}_g, \mathbb{P}_t) = \sup_{\|f\|_L \leq 1} \mathbb{E}_{x \sim \mathbb{P}_g}[f(x)] - \mathbb{E}_{x \sim \mathbb{P}_t}[f(x)], \tag{2}$$

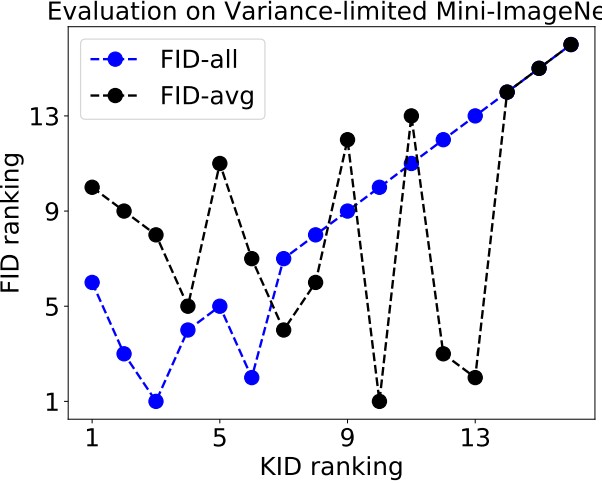

Figure 6: Comparing FID-based and KID-based rankings of variance-limited federated Mini-ImageNet-based simulated generative models. The lower the rank is, the better.

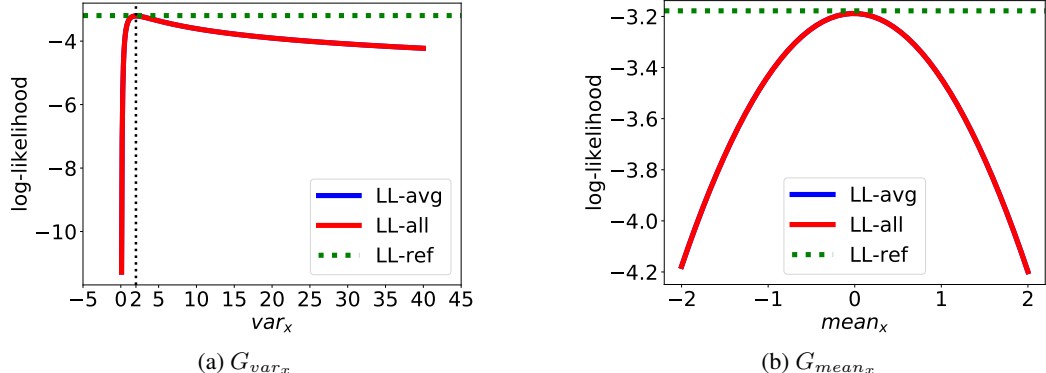

(a) $G_{var_x}$          (b) $G_{mean_x}$

Figure 7: Evaluation of synthetic Gaussian data with the aggregate log-likelihood scores.

where the supremum is over all the 1-Lipschitz functions $f : \mathcal{R}^n \to \mathcal{R}$. Therefore, if we have a parameterized family of functions $\{f_\theta\}_{\theta \in \Theta}$ that a 1-Lipschitz, we could considering solve this problem,

$$\max_{\theta \in \Theta} \mathbb{E}_{x \sim \mathbb{P}_g}[f_\theta(x)] - \mathbb{E}_{x \sim \mathbb{P}_t}[f_\theta(x)]. \tag{3}$$

To estimate the supremum of Equation (2), we employ a family of non-linear neural network $f_\theta$ which are repeatedly stacked by the fully connected layer, the spectral normalization and RELU activation layer. There are three repeated blocks in the network $f_\theta$ and the last block does have RELU. The feature is extracted by pre-trained Inception-V3 network. By optimizing the parameters in $f_\theta$ to maximize $\mathbb{E}_{x \sim \mathbb{P}_g}[f_\theta(x)] - \mathbb{E}_{x \sim \mathbb{P}_t}[f_\theta(x)]$ over $\mathbb{P}_g$ and $\mathbb{P}_t$, we can finally get an estimation of $W(\mathbb{P}_g, \mathbb{P}_t)$. And similarly, we can also define average score W-avg and collective-data-based score W-all under the distributed learning setting. Similar to the CIFAR100 experiment in the main body of paper, we extracted samples from each single class of CIFAR100 and evaluate these samples on federated CIFAR10 dataset. We illustrate a subset of W-avg / W-all pairs in Figure 9. According to experiment results, we find that general 1-Wasserstein-Distance evaluation metric also shows inconsistent behaviours in the distributed evaluation settings.

## 3.7 THE EFFECT OF INTRA-CLIENT VARIANCE

In the main body of this paper, we choose $K = 20$ when we conduct the variance-limited federated CIFAR10 dataset. Hyper-parameter K controls the intra-client variance, the larger the $K$ the larger the

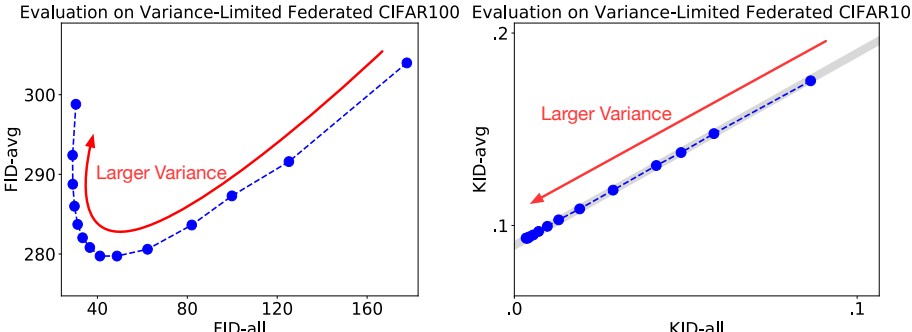

Figure 8: Evaluation on Variance-Limited Federated CIFAR100.

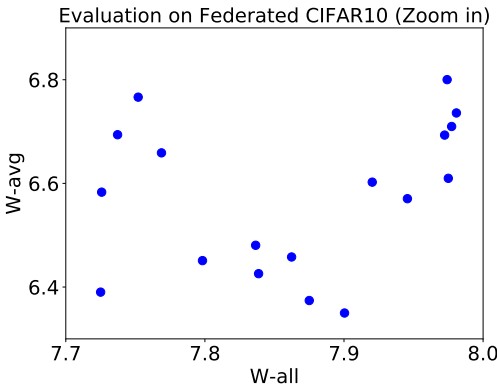

Figure 9: Evaluation with 1-Wasserstein-Distance on Federated CIFAR10.

variance. The number of $K$ will not affect the key conclusion. We prove this claim by conducting an ablation study on hyper-parameter $K$. The $K$ is selected from {5,10,20,50} in our experiment. The results are illustrated in Figure 10. Each of these figure gives a U-shape curve, which indicates that the rankings given by FID-all and FID-avg are highly inconsistent, especially when the intra-client variance and inter-client variance are mismatched.

## 3.8 FULL EVALUATION RESULT OF CIFAR100 ON FEDERATED CIFAR10

We evaluate each class of CIFAR100 on the federated CIFAR10 dataset and report the full results on Table 1.

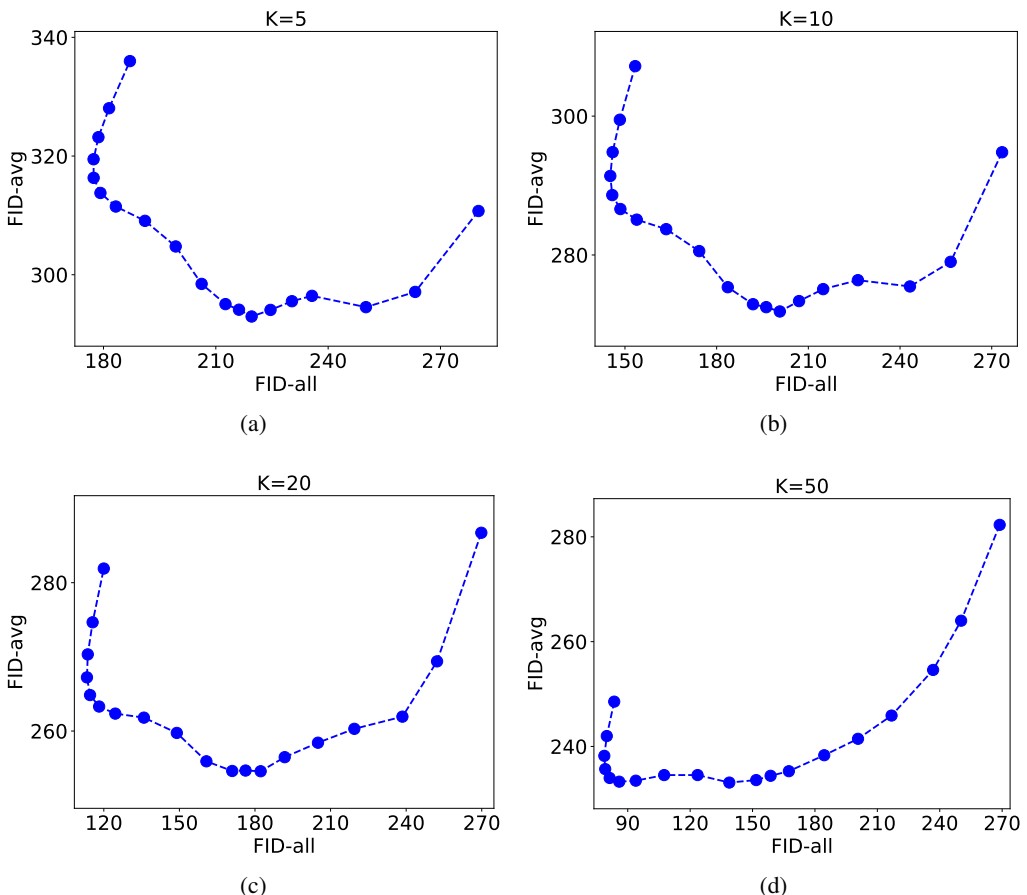

Figure 10: Ablation study on hyper-paramter K.

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

| Class | FID-all | FID-avg | KID-all | KID-avg | Class | FID-all | FID-avg | KID-all | KID-avg |
|---|---|---|---|---|---|---|---|---|---|
| 0 | 267.4 | 285.9 | 0.201 | 0.253 | 25 | 142.4 | 173.2 | 0.067 | 0.116 |
| 1 | 173.2 | 205.5 | 0.109 | 0.165 | 26 | 139.4 | 175.5 | 0.063 | 0.114 |
| 2 | 151.8 | 185.4 | 0.072 | 0.123 | 27 | 114.5 | 153.7 | 0.054 | 0.102 |
| 3 | 124.3 | 162.2 | 0.047 | 0.100 | 28 | 182.0 | 205.0 | 0.112 | 0.160 |
| 4 | 117.0 | 156.1 | 0.049 | 0.100 | 29 | 143.3 | 185.2 | 0.083 | 0.133 |
| 5 | 157.6 | 185.2 | 0.088 | 0.138 | 30 | 147.4 | 179.8 | 0.084 | 0.134 |
| 6 | 142.1 | 179.7 | 0.062 | 0.115 | 31 | 160.4 | 193.6 | 0.108 | 0.158 |
| 7 | 144.9 | 179.2 | 0.066 | 0.113 | 32 | 115.4 | 156.4 | 0.032 | 0.085 |
| 8 | 155.0 | 187.7 | 0.085 | 0.135 | 33 | 146.9 | 181.2 | 0.077 | 0.126 |
| 9 | 180.7 | 203.6 | 0.106 | 0.156 | 34 | 127.5 | 163.4 | 0.068 | 0.117 |
| 10 | 183.2 | 208.8 | 0.099 | 0.151 | 35 | 150.4 | 184.7 | 0.080 | 0.131 |
| 11 | 151.9 | 184.9 | 0.073 | 0.127 | 36 | 142.5 | 176.5 | 0.070 | 0.126 |
| 12 | 126.1 | 163.1 | 0.056 | 0.108 | 37 | 126.7 | 162.9 | 0.066 | 0.118 |
| 13 | 127.0 | 159.4 | 0.058 | 0.112 | 38 | 110.5 | 151.3 | 0.050 | 0.100 |
| 14 | 145.8 | 182.9 | 0.071 | 0.126 | 39 | 233.5 | 257.2 | 0.145 | 0.196 |
| 15 | 124.2 | 165.2 | 0.054 | 0.107 | 40 | 158.9 | 187.6 | 0.078 | 0.128 |
| 16 | 190.3 | 213.6 | 0.112 | 0.161 | 41 | 151.1 | 181.0 | 0.072 | 0.126 |
| 17 | 161.5 | 192.8 | 0.117 | 0.165 | 42 | 137.2 | 174.7 | 0.071 | 0.122 |
| 18 | 142.7 | 179.8 | 0.069 | 0.118 | 43 | 152.0 | 186.0 | 0.090 | 0.139 |
| 19 | 112.4 | 154.1 | 0.047 | 0.101 | 44 | 126.7 | 165.8 | 0.050 | 0.103 |
| 20 | 194.2 | 215.9 | 0.123 | 0.175 | 45 | 135.4 | 173.0 | 0.059 | 0.111 |
| 21 | 164.8 | 194.9 | 0.099 | 0.148 | 46 | 155.1 | 187.1 | 0.080 | 0.133 |
| 22 | 205.1 | 227.0 | 0.126 | 0.180 | 47 | 174.2 | 203.4 | 0.118 | 0.169 |
| 23 | 191.1 | 219.7 | 0.130 | 0.186 | 48 | 145.0 | 175.3 | 0.077 | 0.129 |
| 24 | 174.9 | 202.8 | 0.105 | 0.155 | 49 | 164.8 | 194.6 | 0.110 | 0.164 |
| 50 | 113.8 | 154.0 | 0.045 | 0.095 | 75 | 151.7 | 184.3 | 0.096 | 0.145 |
| 51 | 150.4 | 184.6 | 0.073 | 0.125 | 76 | 151.5 | 183.5 | 0.068 | 0.118 |
| 52 | 195.3 | 222.1 | 0.167 | 0.218 | 77 | 139.0 | 174.9 | 0.066 | 0.117 |
| 53 | 279.7 | 299.1 | 0.217 | 0.270 | 78 | 197.3 | 228.3 | 0.130 | 0.182 |
| 54 | 170.2 | 201.3 | 0.098 | 0.149 | 79 | 138.9 | 174.8 | 0.062 | 0.114 |
| 55 | 104.7 | 146.3 | 0.034 | 0.086 | 80 | 111.1 | 149.9 | 0.043 | 0.095 |
| 56 | 144.2 | 178.5 | 0.075 | 0.126 | 81 | 131.7 | 165.3 | 0.068 | 0.119 |
| 57 | 192.5 | 219.9 | 0.115 | 0.163 | 82 | 197.6 | 227.2 | 0.123 | 0.178 |
| 58 | 131.5 | 161.0 | 0.067 | 0.121 | 83 | 202.6 | 230.6 | 0.122 | 0.173 |
| 59 | 149.7 | 183.7 | 0.093 | 0.144 | 84 | 144.5 | 177.3 | 0.065 | 0.111 |
| 60 | 188.0 | 216.0 | 0.144 | 0.197 | 85 | 123.0 | 160.7 | 0.079 | 0.125 |
| 61 | 249.2 | 270.6 | 0.179 | 0.229 | 86 | 168.0 | 193.0 | 0.083 | 0.135 |
| 62 | 202.1 | 230.6 | 0.133 | 0.184 | 87 | 170.5 | 196.0 | 0.098 | 0.152 |
| 63 | 140.6 | 175.3 | 0.071 | 0.120 | 88 | 133.4 | 170.6 | 0.068 | 0.120 |
| 64 | 118.7 | 156.1 | 0.049 | 0.100 | 89 | 122.3 | 158.1 | 0.059 | 0.112 |
| 65 | 102.2 | 142.2 | 0.024 | 0.077 | 90 | 110.7 | 148.3 | 0.041 | 0.093 |
| 66 | 121.7 | 159.1 | 0.054 | 0.105 | 91 | 124.0 | 160.7 | 0.048 | 0.098 |
| 67 | 132.5 | 167.8 | 0.063 | 0.115 | 92 | 175.4 | 206.2 | 0.096 | 0.149 |
| 68 | 139.7 | 173.1 | 0.073 | 0.123 | 93 | 129.7 | 166.3 | 0.054 | 0.109 |
| 69 | 143.2 | 176.0 | 0.068 | 0.121 | 94 | 213.4 | 235.2 | 0.162 | 0.212 |
| 70 | 178.4 | 209.5 | 0.095 | 0.148 | 95 | 154.7 | 185.2 | 0.084 | 0.133 |
| 71 | 169.4 | 199.0 | 0.120 | 0.167 | 96 | 147.8 | 181.7 | 0.090 | 0.138 |
| 72 | 114.1 | 155.6 | 0.043 | 0.094 | 97 | 137.1 | 171.4 | 0.068 | 0.119 |
| 73 | 137.9 | 170.8 | 0.071 | 0.124 | 98 | 157.1 | 188.6 | 0.082 | 0.134 |
| 74 | 124.7 | 162.1 | 0.061 | 0.108 | 99 | 204.9 | 233.9 | 0.143 | 0.192 |

Table 1: Full evaluation of CIFAR100 on Federated CIFAR10.