# OpenReview forum: "On the Evaluation of Generative Models in Distributed Learning Tasks"
_ICLR.cc/2024/Conference — Submitted to ICLR 2024_

### Official Review · Reviewer_MMnw · 2023-10-25

**Soundness:** 3 good
**Presentation:** 4 excellent
**Contribution:** 2 fair
**Rating:** 6
**Confidence:** 4

**Summary:**

The paper focuses on the evaluation of generative models in the distributed/federated setting and studies two popular metrics, Fréchet Inception Distance (FID) and Kernel Inception Distance (KID). In particular, the authors consider two distributed evaluation modes: (all) the metrics are computed against the global data distribution, i.e., the mixture defined over the clients’ distributions; and (avg) the metrics are computed against the distribution of each individual client and then averaged to define a single score. The authors prove that in the case of KID, both evaluation modes rank generative models in the exact same manner, but that in the case of FID, the rankings defined by FID-avg and FID-all can vary significantly. The theoretical results are further supported by extensive empirical results on popular benchmarks for image generation in a federated setting with heterogeneous client distributions.

**Strengths:**

- To the best of my knowledge, the main results of the paper are novel, and the supporting arguments, both theoretical and empirical, seem to be sound. The evaluation of generative models is an important and challenging topic that becomes even more difficult in the federated setting, which thus far has received little attention in the literature.
- The paper is very well written and easy to read. The authors do a good job of introducing the relevant concepts and literature around generative models and federated learning.
- The experiments are extensive and support the main theoretical claims of the paper. The authors also shared their code, and I have no reason to believe the results are not reproducible.

**Weaknesses:**

- The need for the two modes of evaluation the authors consider (all and avg) is not well motivated in the paper. Among the cited previous works on federated learning for generative models, most if not all rely on some form of FID-all, typically by computing FID scores on a separate test dataset. Are there any examples in the literature where scores similar to FID-avg or KID-avg were considered? I imagine one would be interested in FID-avg in the context of personalization or when privacy concerns prevent the usage of FID-all, for example, but that is never discussed in detail in the current version of the paper.
- Following on the previous point, although the paper is well presented, novel and sound, I am not entirely convinced of its significance.
- The discussion of the results could exploit a few points more in depth. For instance, the authors could comment on how their results could be used to guide the selection of an appropriate metric. From where I stand, it seems the results favor KID-avg, since it can be computed in a distributed manner (thus preserving privacy) but still retains the same ranking of KID-all, which captures the global distribution.

Minor points:
- Theorem 1: “distributions” is misspelled, and I believe “following” should be singular.

**Questions:**

1. In Section 5.1., the authors note that “[…] counterintuitively, the ‘ideal estimator’ did not reach the minimum average of the Fréchet distances”. Could they share any intuition as to why that is the case? Does that indicate taking the average among clients is a poor metric to optimise for?
2. Would it not be possible to also show the optimal value (referent to the true distribution) for the KID metric in the experiments of Section 5.1.?
3. In page 7, the authors comment on the effect of privacy considerations when choosing between FID-all or FID-avg: “[…] a distributed computation of FID-all is more challenging than obtaining FID-avg due to privacy considerations”. This shows an interesting trade-off between the two modes that the authors could explore a bit further. While FID-all can be more challenging to compute from a privacy perspective, wouldn’t FID-avg favor a model that fits the distributions of only one or a few of clients very well, thus potentially “memorizing” the data of these clients?
4. Empirically, does the ranking provided by KID matches any of those given by FDI-all or FDI-avg? It is not clear to me whether we can infer that from the plots, but it would be interesting to know how the rankings of FDI and KID compare in the experiments (even though the theoretical results have nothing to say here).
5. On a similar note, for the experiments of Section 5.1. as well as those with DDPM, one could compute (estimates of) the log-likelihood of the data. Have the authors considered how the ranking defined by the log-likelihood compare with the other metrics? The log-likelihood should provide consistent rankings, differently from the density metric of Naeem et al.

---

> ### Author Response · Authors · 2023-11-20
> **Authors' Response to Reviewer MMnw**
>
> We thank Reviewer MMnw for his/her time and constructive feedback and suggestions. The following is our response to the comments and questions in the review.
>
> **1- Motivation of our study of the evaluation score aggregations.**
>
> **Re:** As pointed out by the reviewer, the existing literature on federated generative model learning mostly utilize the distribution-level aggregate scores (i.e. FID-all and KID-all). However, in a real federated learning setting, the computation of the FID-all and KID-all scores could be highly challenging due to common privacy and computation constraints in federated learning tasks. For the distributed computation of both KID-all and FID-all, the clients should either share their entire dataset with the server, which would violate the privacy constraints, or solve a completely independent distributed optimization problem which incur great communication and computation costs. Under such constraints, the computation of the averaged score of individual clients (FID-avg and KID-avg) will be significantly less expensive and thus more feasible.
>
> In our work, we show that although KID-all and KID-avg will take different values, they will result in the same ranking of generative models (Theorem 2). This result reveals an important advantage of KID evaluation in distributed settings, since to obtain the models' KID-all ranking the clients need to share only their individual KID scores with the server. On the other hand, we show that the same ranking consistency does not hold for FID-avg and FID-all. We characterize the gap between the two aggregate scores which depend on the choice of generative model $P_G$. Based on the reviewer’s comments and writing suggestions, we have added one more paragraph to the introduction explaining the above motivation of our study.
>
>
> **2- The ideal estimator in Section 5.1**
>
> **Re:** We note that FD-avg ranks the models in terms of the average Frechet distance with respect to the two Gaussian modes $\mathcal{N}_1,\mathcal{N_2}$ representing the clients. Therefore, the ideal estimator is a mixture distribution $\frac{1}{2}(\mathcal{N}_1+\mathcal{N_2})$ combining the two modes. However, the Frechet distance of the mixture distribution to each of the modes $\mathcal{N}_1,\mathcal{N_2}$ is non-zero, and therefore it is not clear if the mixture distribution can minimize the sum of the Frechet distances. Our experiments show that the ideal estimator’s mixture distribution $\frac{1}{2}(\mathcal{N}_1+\mathcal{N_2})$ is indeed *not the minimizer* of the sum of Frechet distances to $\mathcal{N}_1$ and $\mathcal{N}_2$.
>
> **3- KD-ref in Figure 1.b**
>
> **Re:** We thank the reviewer for pointing out the lack of KD-ref line in Figure 1.b. We have added the line for KD-ref (which equals 3.79) in the updated figure 1b.
>
> **4-“Wouldn’t FID-avg favor a model that fits the distributions of only one or a few of clients very well?"**
>
> **Re:** As shown in Theorem 1, the optimal mean vectors (in the InceptionNet-based embedding) for FID-all and FID-avg are identical and equal to the averaged mean of clients: $\widehat{\mu}=\sum_{i=1}^k \lambda_i\mu_i$. Please note that $\widehat{\mu}$ is the mean of the collective distribution across clients, and therefore the optimal solution based on FID-avg could not fit only a few clients’ distributions and miss the majority of clients’ distributions.
>
> **5- Empirical analysis of the relationship between FID-based and KID-based rankings**
>
> **Re:**  In the updated supplementary material (Section 3.3 and Figures 2,3,4), we perform an experiment on the $1024\times1024$ FFHQ dataset using the StyleGAN2 generator. The rankings based on FID-avg and KID (note that KID-all and KID-avg give the same ranking) were highly inconsistent (Figures 5,6). On the other hand, as shown in Figures 5,6, the rankings given by FID-all and KID (both KID-all and KID-avg) were more consistent, showing KID-avg-based rankings could correlate well with the FID-all-based rankings.
>
> **6-Log-likelihood-based evaluation of the experiment with synthetic data**
>
> **Re:** Based on the reviewer’s suggestion, we computed the averaged log-likelihood (LL) scores for the synthetic Gaussian generators. Please note that LL-avg and LL-all take the same value since the log-likelihood score is the expected value of the sample-based score at every client. We observed that the averaged log-likelihood score is optimized at nearly the same point as FD-all and KD-all, showing that the solution found by FD-all and KD-all are consistent with the maximum likelihood solution. The results of the numerical experiments can be found in Section 3.4 (Figure 7) of the revised Appendix. We note we could not perform the log-likelihood-based evaluation of the DDPM model, since we cannot compute the PDF of the DDPM generative model needed for the computation of the log likelihood scores.

---

> ### Comment · Reviewer_MMnw · 2023-11-21
>
> I thank the authors for their answers as well as the new additions to the paper. The updated version of the paper is more clearly motivated and the extra experimental results provide new relevant insights into how FID-avg and KID-avg behave in practice.

---

> > ### Author Response · Authors · 2023-11-23
> > **Thank you for your feedback and suggestions**
> >
> > We thank Reviewer MMnw for his/her time, constructive suggestions in the review, and positive feedback on our response.

---

### Official Review · Reviewer_1nBR · 2023-10-31

**Soundness:** 2 fair
**Presentation:** 3 good
**Contribution:** 2 fair
**Rating:** 5
**Confidence:** 3

**Summary:**

This paper studied the evaluation of generative models in distributed learning settings, in particular, the federated learning scenario. The paper showed that in distributed settings, the way to aggregate evaluation metrics may affect rankings of generative models. For FID score, the paper theoretically showed that FID-avg which is the mean of clients’ individual FIDs, can be inconsistent with FID-all, which is the FID score computed on the collective dataset, leading to different model rankings. On the other hand, for another evaluation metric KID (kernel inception distance), KID-avg and KID-all are always consistent for ranking models. Experimental results were provided to support the theoretical findings.

**Strengths:**

1. The paper provided theoretical findings on evaluation metrics for generative models in distributed settings. The results could be of interest and are worth discussing when training and comparing generative models in the distributed manner.

2. The paper provided experimental results on toy datasets and real image datasets, to show that while FID scores can be inconsistent, the KID scores are always consistent.

**Weaknesses:**

1. For FID scores, while the paper gave theoretical formulations for FID-avg and FID-all, the formulations can only distinguish different rankings of generators based on their distances to corresponding covariance matrices. It would be more informative to characterize the gap between FID-avg and FID-all with the distances between clients’ data distributions. This could allow one to estimate the degree of FID score inconsistency based on how different clients’ datasets are.

2. In experiments, it seems that most results for FID scores come from ``simulated’’ generators, that is, treating a class of images from CIFAR-10 or CIFAR-100 as the output of a generator. Such simulation does not well represent real generators trained on the multi-class dataset, or the general federated learning protocol (even when each client owns one class of images only). Hence the results do not sufficiently reveal if inconsistency between FID-avg and FID-all is an actual issue in practice. Moreover, the FID scores are quite large and far from optimal, in which case the consistency of the evaluation metric may not be a most important property to pursuit.

**Questions:**

In Figure 2, do the KID scores imply that the plane generator is better than the DDPM model, while the former generates much less diverse images? This may raise a question of how to properly evaluate the models apart from consistency.

---

> ### Author Response · Authors · 2023-11-20
> **Authors' Response to Reviewer 1nBR**
>
> We thank the reviewer for his/her time and constructive feedback on our work. Here is our response to the comments and questions in the review.
>
> **1- The gap between FID-avg and FID-all**
>
> **Re:** We note that Theorem 1 characterizes the gap between FID-all and FID-avg up to a $\mathrm{constant}_1$ (with respect to generative model $P_G$), as it shows
>
> $$FID_{\mathrm{avg}} - \mathrm{FID}_{\mathrm{all}} =\mathrm{FID}(\widetilde{P}_X;P_G) - \mathrm{FID}(\widehat{P}_X;P_G) + \mathrm{constant}_1$$
>
> Here $\widehat{P}_X$ is the normal distribution with the aggergate mean $\widehat{\mu} = \sum_i \lambda_i \mu_i$ and aggregate covariance matrix $\widehat{C} = \sum_i \lambda_i (C_i + \mu_i\mu_i^\top - \widehat{\mu}\widehat{\mu}^\top )$, and $\widetilde{P}_X$ is the Barycenter normal distribution with the same mean $\widehat{\mu} = \sum_i \lambda_i \mu_i$ and barycenter covaraince matrix $\widetilde{C}$ defined in Theorem 1 (aggergating only $C_1,\ldots, C_k$). Therefore, the gap between FID-avg and FID-all gap can be simplified to the following for a $\mathrm{constant}_2$ (with respect to generative model $P_G$)
>
> $$FID_{\mathrm{avg}} - \mathrm{FID}_{\mathrm{all}}= 2\mathrm{Tr}((C_G\widehat{C})^{1/2}- (C_G\widetilde{C})^{1/2} ) + \mathrm{constant}_2$$
>
> As can be seen, the gap between FID-all and FID-avg depends on the choice of $C_G$ (generative model's covariance matrix) and hence is not independent of $P_G$. We have included this discussion in the revised Remark 3.
>
> **2- Using simulated generators in the experiments**
>
> **Re:** We note that our main goal in the numerical experiments is to show the potential inconsistencies between FID-avg and FID-all ranking and contrast it with the KID-avg and KID-all consistent rankings. As implied by the theoretical and numerical results, a heterogeneous distributed setting with a significantly smaller intra-client variance than inter-client variance could lead to inconsistent  FID-based rankings of generative models. This scenario could potentially occur in real-world federated learning tasks, because the samples within each client are expected to have a considerably lower variance than the collective set of all clients’ samples.
>
> To address the reviewer's comment on the lack of real generators in our experiments, we performed an experiment using a pre-trained StyleGAN generator. In the experiments, we simulated a heterogeneous setting by modeling the client distributions as truncated normal distribution for the GAN's latent vector with differently selected mean vectors at different clients. We then evaluated the FID and KID aggregated scores for the evaluation of a truncated StyleGAN generator with different truncation coefficients, which is a similar experimental setting to the related work [1].  Our numerical results in the Appendix showed the same consistency trends that we reported for the simulated generators in the main text. We visualize the client-based samples and report the aggregated scores in the updated supplementary document (Figures 2,3,4).
>
> **3- The KID-based Ranking of Models in Figure 2**
>
> **Re:** As pointed out by the reviewer, the KID-score with polynomial kernel prefers the less-diverse simulated plane generator. Please note that the KID-based ranking of generative models highly depends on the choice of kernel similarity function $k(\cdot,\cdot)$ in the KID definition. In fact, we numerically validated that a different choice of kernel function (Gaussian kernel) will prefer the more-diverse DDPM generated samples. We discussed the numerical comparison with different kernel functions in Section 3.3 in the updated supplementary material.
>
> Finally, we note that our result in Theorem 2 on the consistent ranking of KID-all and KID-avg applies to every choice of kernel function $k$. However, the study of the proper choice of kernel functions in different machine learning applications is beyond the scope of our work, and could be an interesting research direction for future exploration.
>
> [1] Kynkäänniemi, Tuomas, et al. "Improved precision and recall metric for assessing generative models." Advances in Neural Information Processing Systems, 2019.

---

### Official Review · Reviewer_dkjU · 2023-11-07

**Soundness:** 3 good
**Presentation:** 3 good
**Contribution:** 1 poor
**Rating:** 3
**Confidence:** 3

**Summary:**

This paper explores the extension of FID and KID scores from centralized setting to distributed setting. Average of scores at each client is compared to the corresponding centralized score. Authors prove that the FID score rankings may not match, while KID scores always do. Experiments confirm this theoretical claim.

**Strengths:**

- Important setting: Distributed learning and evaluation of generative models is an increasingly important topic
- The theoretical analysis is accurate and empirical results adequately support the theory

**Weaknesses:**

- The paper’s main contribution is a very simple observation. In essence, the FID score is an infimum of an expected distance, whereas KID is expectation of the kernel distance. Hence, KID ranking consistency directly follows from linearity of expectation, while FID does not follow the same. This observation alone does not constitute a significant contribution.
- The simple observation would’ve still constituted as a good contribution if the authors provide some actionable insights. For example, one possible conclusion could be that one should use KID rather than FID in distributed settings — however, authors do not compare models obtained via these two methods in detail. Another possible direction could be to modify FID so that the new score behaves well under averaging.

**Questions:**

N/A

---

> ### Author Response · Authors · 2023-11-20
> **Authors' Response to Reviewer dkjU**
>
> We thank Reviewer dkjU for his/her time and feedback. In the following, we respond to the reviewer’s comments on our results:
>
> **1- “KID is the expectation of the kernel distance. Hence, KID ranking consistency directly follows from linearity of expectation”**
>
> **Re:** We would like to respectfully point out the reviewer’s misunderstanding of linearity properties of KID distance. We think the reviewer’s comment is due to a misinterpretation of the equation defining KID at the beginning of Page 4. Please note that the first term in that definition of KID, i.e. $\mathbb{E}_{X,X’\sim p_X}[k(X,X’)]$, is not a linear function of $P_X$. Although expectation is a linear operator, this expected value is over two independent samples $X,X’$ distributed according to $P$, which makes the expectation and hence $\mathrm{KID}(P,Q)$ a non-linear function of $P$. To see this, please note
>
> $$\mathbb{E}_{X,X’\sim p_X}\bigl[ k(X,X’) \bigr] = \underset{\text{non-linear in $p_X$}}{\underbrace{\int p_X(x)p_X(x’) k(x,x’) \mathrm{d}x\mathrm{d}x’}}$$
>
> In general, for a fixed $Q$, the KID distance $\mathrm{KID}(P,Q)$ is not a linear function of $P$ and hence cannot be the expectation of a random distance according to distribution $P$. To see this more clearly, we refer to the *strictly convex* property of the KID function which holds for every universal kernel, e.g. Gaussian kernel, as shown in [1]. Note that the strict convexity of KID implies that for every distributions $Q, P_1\neq P_2$:
>
> $$\mathrm{KID}\bigl(\frac{1}{2}P_1 + \frac{1}{2}P_2  ,  Q  \bigr) \neq  \frac{1}{2} \mathrm{KID}(P_1,Q) + \frac{1}{2} \mathrm{KID}(P_2,Q)$$
>
> This nonlinearity property is also evident in Theorem 2, showing the averaged $\text{\rm KID-avg}$ is *strictly greater* than $\text{\rm KID-all}$. If Theorem 2 simply followed from the linearity of expectation, as stated in the review, $\text{\rm KID-avg}$ would have been equal to $\text{\rm KID-all}$ which is not the case according to Theorem 2.
>
> **2- “the FID score is an infimum of an expected distance… Hence, FID does not follow the same”**
>
> **Re:** We respectfully disagree with the reviewer that our result on the inconsistent ranking based on FID-avg and FID-all is only a consequence of the FID score being the infimum of an expectation. To show why the infimum-based definition of FID is not enough to imply the inconsistent ranking, we refer to Theorem 2 on the consistent ranking between KID-avg and KID-all for the KID metric. Please note that, as discussed in [2],  KID is similarly the supremum of an expectation:
>
> $$\mathrm{KID}(P,Q) := \sup_{\phi: \Vert \phi\Vert_{H_k}\le 1} E_{X\sim P}[\phi(X)] -  E_{X’\sim Q}[\phi(X’)]$$
>
> If our results on the inconsistent ranking of FID aggregations only followed from the infimum-based definition of FID, the same inconsistent ranking would have also been the case for KID because KID is also the supremum of an expectation. However, Theorem 2 in the paper suggests that the opposite is the case for KID, and KID-all and KID-avg result in the same ranking of generative models while they take different values.
>
> We hope the above explanations help with the reviewer’s concerns on the significance of our theoretical contribution and will be happy to further discuss any remaining concerns on the significance of our theoretical results.
>
> **3- Our results and the choice of evaluation metric in distributed settings**
>
> **Re:** A practical implication of our results is the lower privacy and computational costs of ranking generative models in the distributed settings using the KID metric. This is because instead of computing KID-all which requires sharing massive information on the clients' data, we can use KID-avg (requiring each client to only share their KID score), which, as Theorem 2 shows leads, gives the same ranking of generative models. However, the privacy and computational expenses of an FID-based evaluation could be significantly higher, since averaging the clients’ individual FID scores could lead to an inconsistent ranking of the models with the original FID-all metric.
>
> [1] Gretton, et al, "A kernel two-sample test." The Journal of Machine Learning Research (JMLR), 2012. \
> [2] Bińkowski, et al, "Demystifying MMD GANs." In International Conference on Learning Representations (ICLR), 2018

---

### Meta-Review · Area_Chair_qDxx · 2023-12-14

**Metareview:**

This paper did not have enough support from the reviewers to warrant publication at ICLR. While some of the concerns from the reviewers were addressed in the rebuttal (in particular, clarifying a misunderstanding about the non-linearity in P_X of the KID), two out of three reviewers still thought the paper would require a major revision, in particular, including additional non-synthetic experiments.

**Justification For Why Not Higher Score:**

Not enough support from the reviewers.

**Justification For Why Not Lower Score:**

N/A

---

### Decision · Program_Chairs · 2024-01-16

Reject